# Detecting Emerging Symptoms of COVID-19 using Context-based Twitter Embeddings

**Roshan Santosh[1], H. Andrew Schwartz[2], Johannes Eichstaedt[3],**
**Lyle Ungar[1], Sharath Chandra Guntuku[1]**
[1]University of Pennsylvania [2]Stonybrook University [3]Stanford University
{roshansk,sharathg}@cis.upenn.edu

## Abstract

In this paper, we present an iterative graph-based approach for the detection of symptoms of COVID-19, the pathology of which seems to be evolving. More generally, the method can be applied to finding context-specific words and texts (e.g. symptom mentions) in large imbalanced corpora (e.g. all tweets mentioning #COVID-19). Given the novelty of COVID-19, we also test if the proposed approach generalizes to the problem of detecting Adverse Drug Reaction (ADR). We find that the approach applied to Twitter data can detect symptom mentions substantially before being reported by the Centers for Disease Control (CDC).

## 1 Introduction

The COVID-19 pandemic has interrupted many everyday behaviors. SARS-CoV-2 is a relatively new virus and gaps in knowledge persist about how it affects the body, and consequently, its symptoms and symptom severity. In the early phases of the pandemic, patients and providers in affected areas used social media to exchange information about symptoms and clinical treatment (Iacobucci, 2020; Stokes et al., 2020). During COVID-19, the use of social media has increased dramatically (>20%) as individuals shelter in place (Venkatraman, 2020). While social media is non-representative and contains misinformation (Singh et al., 2020), it provides an open forum for individuals to share their perceptions, concerns (Giorgi et al., 2020), understanding of health and science (Yang et al., 2018), and mental health status during emergencies when wide spread polling may not be available (Guntuku et al., 2020).

Social media could enable early symptom discovery for diseases such as COVID-19 where the pathology is not completely known and our knowledge of it is evolving (Del Rio and Malani, 2020). The most prominent symptoms such as fever, cough, and shortness of breath were known early on during the COVID-19 pandemic. However, others such as changes in smell/taste, body aches, and diarrhea were added later to the symptom list by the CDC (Grant et al., 2020).

Using social media to gather information on public health is a growing focus of research, with a special emphasis on discovering side effects of drugs (pharmacovigilance) (O'Connor et al., 2014), often using labeled datasets to build supervised machine learning models (Luo et al., 2017).

We propose a graph-based natural language processing framework to automatically detect emerging symptoms using Twitter data. Our approach is built on the hypothesis that by identifying token embeddings that capture the context of symptom mentions, new tokens used in a similar context can be identified through embedding similarity (Devlin et al., 2018). Our approach shares similarities with the idea of lexicon development (Bontcheva et al., 2013), which uses an unsupervised graph-based approach for labeling new words given a few labeled words. However, the graph is initiated with words of interest that have already been identified.

Our method's focus on a specific context allows it to search through large imbalanced corpora to identify context-specific (e.g. symptoms) tweets. This differentiates it from previous works by (Wu et al., 2019; Mpouli et al., 2020) that identify domain specific lexicon. Further, the approach by (Wu et al., 2019) relies on a domain specific corpus and topic modeling to build a lexicon, which would require the construction of a symptom-specific COVID-19 corpus.

## 2 Method

As is the case with several applications involving creating word lists associated with a construct

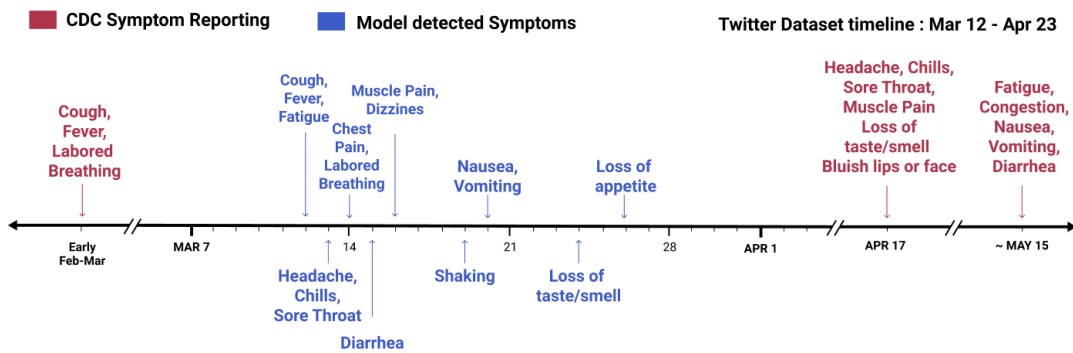

Figure 1: Comparative timeline of symptom detection by our approach against CDC reporting

or topic (Das and Smith, 2012), symptom mentions associated with COVID-19 come in different forms and shapes - often difficult to curate in advance (Rúa, 2007). The approach we propose assumes that we know at least one word of interest (i.e., a seed word) along with few corresponding seed texts where the seed word has been used in the desired context. For the case of emerging symptom detection, *cough*, a seed text could be '*I have a dry cough, chest pain and feeling lethargic as hell plus a headache*'.

## 2.1 Manual Context-Text Approach

Given the seed word and corresponding seed texts, BERT (*bert-base*) embeddings (Devlin et al., 2018) for the seed word are extracted from each of the texts. The BERT embedding for each token was computed by summing the hidden states of the last 4 layers of BERT. Individual embeddings from each of the seed texts are then averaged to generate a representative embedding for the seed word. We use 5 seed texts that capture part of the considerable variance associated with the symptom context.

Using the representative embedding for the seed word, an exhaustive search is performed across the dataset at a token level to identify tokens that are most similar to the seed word, where similarity is measured using cosine similarity (one minus cosine distance). All tokens with a similarity value less than a minimum threshold (set empirically at 0.3) are excluded. Similarity scores of all occurrences of a given word are averaged.

**Training Details** We fine-tuned a *bert-base-uncased* model using the masked language modeling protocol with the following parameters: Num. Epochs: 5, Learing Rate: 1e-5, Batch Size: 8, Max Sequence Length: 128. Embeddings were extracted from the model by summing the hid-

den states of the last 4 BERT layers, resulting in 768-dim embeddings. HuggingFace library (Wolf et al., 2019) was used for loading pre-trained model weights as well as Language Model training.

## 2.2 Graph-based Iterative Training Approach

The previous model required text for every new seed word and did not allow multiple runs with different seeds to learn from each other. To address this, we propose an iterative search model to develop a similarity-based word graph, retaining the search methodology of the earlier approach, but including a graph element and a trainable search parameter that improves the detection of context-specific words with increased iterations.

The directed and weighted word graph of the model represents connections between tokens. Each node in the graph corresponds to a word and is characterized by the representative embedding of the word. The edges have weights corresponding to the similarity score between the connected words (nodes). The second component of the model is the so-called 'Context Embedding', $CEmb$. The context embedding is conceptualized to be an embedding vector that represents the specific context that we are interested in. Initialized by the representative embedding of the seed word, the context embedding incorporates embeddings from other words over iterations, to develop into a more robust representation of the specified context. [1]

## 2.3 Algorithm

**Initialization** Initialize graph **G** by setting the root node with the representative embedding of the seed word. Initialize a queue **Q** by adding the seed

---

[1]Code available at `https://github.com/rsk2327/Covid-Symptoms-NLP`.

word to it. The context embedding **CEmb** is also initialized to the representative embedding of the seed word. **CEmb** $\leftarrow Emb\{$Seed word$\}$, where $Emb\{x\}$ denotes the representative embedding of token $x$.

**Procedure** The specific steps used in the algorithm are as follows:

1. Pop next word from **Q**, denoted by $t$. Initialise a new node in **G** corresponding to $t$ and set the node embedding to $Emb\{t\}$.

2. Initialise the query embedding $q$ as $q \leftarrow k * $ **CEmb** $+ (1 - k) * Emb\{t\}$.

3. Iterate through all tokens in the data, comparing their embeddings against the query embedding $q$. All tokens with similarity less than the minimum similarity threshold $minSimThresh$ are dropped.

4. Select the top $n$ words based on their similarity to $q$. Add these words to **Q**. Instantiate new nodes (if one does not already exist) for these words in **G** and add outgoing edges from $t$ to these new nodes.

5. If all words for given depth are explored, the top $m$ words corresponding to that depth are selected based on similarity to **CEmb**. The context embedding is then updated by averaging $CEmb$ with the representative embeddings of the selected words (Eq 1).

$$\mathbf{CEmb} \leftarrow \frac{\mathbf{CEmb} + \sum_{i=0}^{m} \mathrm{Emb}\{x_i\}}{m + 1} \quad (1)$$

6. Stop iterations when either **Q** is empty or when the maximum depth $maxDepth$ of **G** is achieved. Otherwise, repeat from Step 1.

## 3 Experiments

### 3.1 Manual Context-Text Approach

We tested our approach on a Twitter dataset containing tweets related to COVID-19, collected between March 12 to April 23. Our experiments were run on a random subset of this dataset containing 1 Million tweets.

Our objective for this dataset is to identify new symptoms of COVID-19 mentioned in tweets. We ran tests with *cough*, an established symptom, as the seed word. Seed texts (tweets) were selected where *cough* was used as a symptom to ensure that the correct context is captured. Top 10 results are shown in Table 1. The scores represent the cosine similarity scores of the context embedding of cough with the averaged embeddings of

Table 1: Words returned (and their cosine similarity scores) by Manual Approach described in Section 2.1 and Graph-based Iterative Approach described in Section 2.2

| Seed : Cough (Manual Approach) | | Seed : Cough (Graph Approach) | |
|---|---|---|---|
| **Word** | **Sim.** | **Word** | **Sim.** |
| fever | 0.67 | vomiting | 0.84 |
| throat | 0.62 | fever | 0.83 |
| ##tis (pneumonitis) | 0.61 | throat | 0.82 |
| headache | 0.61 | congestion | 0.75 |
| nose | 0.59 | headache | 0.71 |
| breathing | 0.58 | coughing | 0.70 |
| congestion | 0.57 | asthma | 0.69 |
| ##itis (bronchitis) | 0.57 | ##itis (bronchitis) | 0.64 |
| taste | 0.55 | nausea | 0.63 |
| ##raine (migraine) | 0.54 | ##hea (diarrhea) | 0.45 |

the detected words. With just a single seed word and corresponding text as input, the model could identify key symptoms of COVID.

### 3.2 Graph-based Iterative Training Approach

We evaluate our graph-based approach on 2 different datasets, with each dataset having a different context - 1) COVID-19 Symptom Detection; and 2) Adverse Drug Reaction Identification.

#### 3.2.1 COVID-19 Symptom Detection

With the 1 million COVID-19 tweet dataset, we use *cough* as the seed word, $k = 0.3$, $maxDepth = 3$ and $n = 5$ to test our approach. The resulting graph from our model is shown in Figure 2. The size of the nodes represent the number of occurrences of a token as a symptom while the color intensity of the nodes represent the similarity values computed for the node during the graph building process (shown in Appendix).

We observe that the model identified a wide range of symptoms ranging from common symptoms like fever, fatigue to less common ones like headache, vomiting, (chest) congestion, nausea, mig-##raine.

**Evaluation** Though a quantitative evaluation of our approach is not straightforward as there is no definitive 'ground truth', we evaluate our approach by computing the precision in detecting correct words that fit the specified context.

For the problem of symptom detection, precision is calculated as the percentage of the actual symptoms detected by our model. Given that our model outputs a ranked list of words, precision is computed by looking at the top $p$ results, where $p$

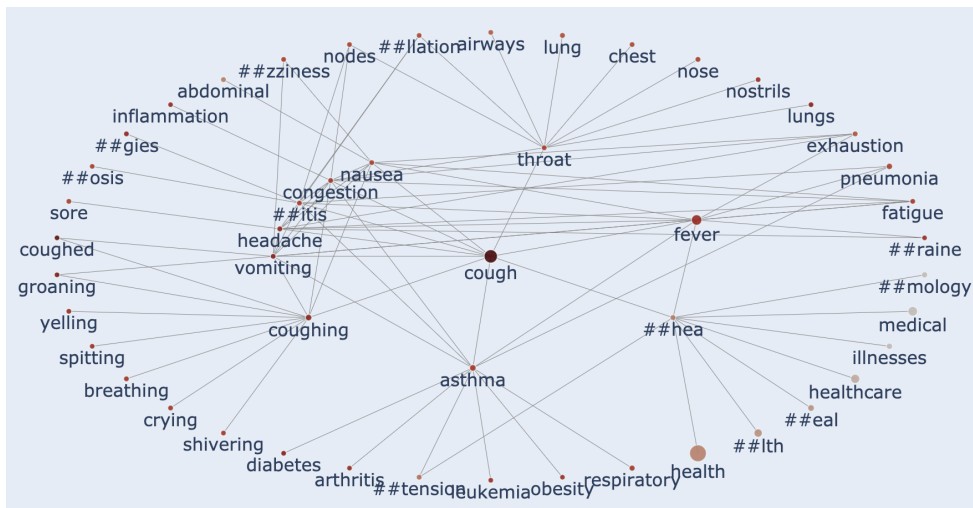

Figure 2: Symptom model graph for COVID-19 Tweet dataset

represents the threshold for computing precision (Table 2).

Table 2: Precision for Symptom Detection

| Model | Seed word | Precision | | |
|---|---|---|---|---|
| | | p = 5 | p = 10 | p = 20 |
| Manual | cough | 0.8 | 0.9 | 0.8 |
| Manual | fever | 1.0 | 0.9 | 0.75 |
| Manual | fatigue | 0.8 | 0.8 | 0.75 |
| Graph | cough | 1.0 | 0.9 | 0.9 |

Through a manual inspection of the top 100 results, rare to-be-confirmed symptoms like lack of appetite, skin/eye irritation, vertigo, anemia were detected. This marks a key utility of our approach as it helps generate potential symptom candidates which can guide further evaluation.

### 3.2.2 Adverse Drug Reaction (ADR) Detection

For the second task, we use an annotated ADR dataset (Sarker and Gonzalez, 2015), where 13% of the tweets are labeled as ADR. The objective of this task is the identification of words denoting adverse drug reactions. Therefore, the specific context of interest is different from the previous dataset where it was identification of symptoms of a disease. By testing our model on this dataset, we also test the ability of our approach to generalize to new tasks.

For the experiment, the seed word used is *pain*. $k = 0.2$, $maxDepth = 3$, and $n = 5$. The resulting graph from our model is shown in Figure 3.

Some of the key ADR identified include inflammation, bleeding, muscle (pain), (skin) lesions, tremors, discomfort, and (calcium) deposits.

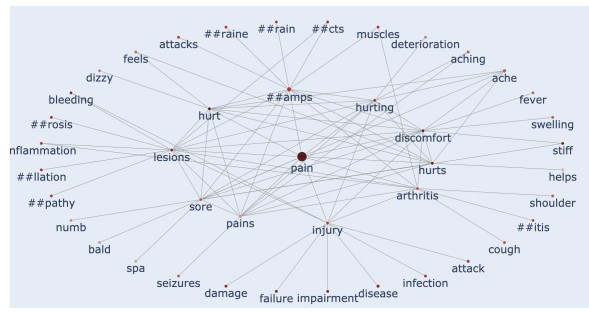

Figure 3: ADR model graph for Statins dataset

**Evaluation**   Similar to COVID-19 symptom detection evaluation, we evaluate the model's performance for ADR detection, where a positive word represents an adverse reaction to a drug (Table 3).

Table 3: Precision for ADR Detection

| Model | Seed word | Precision | | |
|---|---|---|---|---|
| | | p = 5 | p = 10 | p = 20 |
| Manual | pain | 1.0 | 0.9 | 0.8 |
| Graph | pain | 1.0 | 0.9 | 0.95 |

## 4   Discussion

The COVID-19 pandemic evolved in a global climate of confusion and uncertainty. The professional and lay public alike speculated on disease course, severity, and symptoms. The COVID-19 symptoms first observed appeared to be largely non-specific to COVID-19 (e.g., cough, fever). Finding COVID-specific symptoms (such as the sudden loss of the sense of smell) is important and potentially of clinical significance as large populations are being risk-assessed (Wu and McGoogan, 2020). The "digital exhaust" of social media encodes in-

formal case reports of symptoms and discussions of media content about the virus alike. In principle, it could allow for the generation of a "master list" of COVID-19 symptom candidates, which the public health and medical community can, in turn, consider for further evaluation as COVID-specific markers (Chan et al., 2020).

In this study, we present an iterative learning approach to generate such a "master" list of COVID-19 symptoms, using the identification of words matching a specific symptom context. Our approach is intended to work at the intersection of human-machine interaction and provide a ranked list of most important keywords for a user to explore and validate. A typical workflow would involve users from the medical/ pharmacovigilance domain inputting a small number of known keywords of interest, detecting the emerging signals (symptoms/ADR) output by the model, and validating them using the corresponding text. This allows for very early detection of symptoms/ADR from social media platforms without the need to manually scavenge large corpora.

Our initial experiments with Word2Vec (Mikolov et al., 2017), Glove (Pennington et al., 2014), and latent Dirichlet allocation (Blei et al., 2003) for the same task, did not perform well since the similarity between tokens was the same irrespective of the context in which the word was being used. So with 'cough' as the seed word, highly correlated words like 'fever' were detected by these approaches, while words like 'congestion', 'nausea', 'diarrhea' were not. Consequently, we developed an iterative graph-based training approach using BERT embeddings that are able to take into account the context in which words are used.

Through a preliminary evaluation, our approach shows high sensitivity in detected words. The approach detected headache, chills, sore throat, diarrhea, and other symptoms around a month before the CDC reported them (Figure 1). Given the novelty of COVID-19, the current method is hard to evaluate. Recall and other metrics that depend on false negatives are not useful as there is no definitive 'ground truth'. This is even more true for COVID-19 as the symptoms are evolving and newer symptoms are being observed even after several months from the first case. We, therefore, considered the approach in the more studied context of detecting adverse drug reactions and show

that the approach generalizes to this domain. The annotations in ADR vary from very specific entities like 'pain', 'headache' to very broad phrases like 'a tingling sensation' (Nikfarjam et al., 2019). Though our approach does not extract multi-word terms, given that the typical workflow is expected to focus on the validation of symptoms using their corresponding full-length texts, the utility of multi-word terms is minimal.

The approach relies on the Context Embedding contribution parameter ($k$). By varying $k$, we observe a phenomenon analogous to 'Exploration vs Exploitation' (Coggan, 2004), which, in principle, means that this method can be calibrated for different use cases. In the early phase of the disease, for example, a low $k$ parameter may be chosen to aid in the generation of symptom candidates to be considered in light of the emerging clinical literature on COVID-19 and known physiological and biological interactions in the human body. A high $k$ may be chosen to yield the subset of COVID-19 symptoms that are more robustly associated with the disease, at the cost of missing infrequent (albeit potentially specific) disease markers.

In summary, this study demonstrates that the digital traces of social media can be mined effectively to detect emerging symptoms in the population during a public health emergency.

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

# Appendix

**COVID-19 Keywords**    used to pull tweets from the streaming Twitter API: coronavirus, covid, 2019-ncov, covd, outbreak, pandemic, corona, corono, washyourhand, handwashing, stayhome, stayathome, and sarscov.

## Illustrative graph building example

**MODEL INPUT**
**Seed Word** ← cough
**Seed Text** ← [Text A, Text B]
Text A: "I haven't felt "good" for a couple of days. Today I have a headache, cough, chills, fatigued, and feel congested."
Text B: "COVID-19 can induce intense fatigue and trigger a recurring cough and intermittent fever. It varies from person to person"

---

**ITERATION 1**
> Top word (cough) is popped from **Q** and processed* using user-provided seed text
*Processed implies running the similarity search approach where new words are scored based on their cosine distance to the Context Embedding (**CEmb**)

**Output**
- fever, [Text C, Text D]
- headache, [Text E, Text F]
- throat, [Text G, Text H]

*Only one example text per symptom is shown, for brevity*
Text C: "Due to #COVID19 concerns, all visitors are being screened for symptoms such as cough, runny nose, fever and difficulty breathing"
Text E: "I have a dry cough, chest pain and feeling lethargic as hell plus a headache. Not to mention feeling like I have to take deep breaths. Went to urgent care and they basically wrote it off as a UPI because COVID-19 tests are unavailable."
Text G: "I'm still really sick. Been coughing so much my ribs are sore. Have a sore throat, am achy, congested, dizzy, nauseated."

> fever, headache and throat are enqueued to **Q**
> **CEmb** is updated

---

**ITERATION 2**
> Top word (fever) is popped from **Q** and processed. Text corresponding to 'fever' identified from previous iteration (Text C, D) is used as seed text**
**This allows for the automatic evaluation of new seed words without the need for manual seed text input

**Output**
taste, [Text I, Text J]
breathing, [Text K, Text L]
congestion, [Text M, Text N]

*Only one example text per symptom is shown, for brevity*
Text I: "Pretty sure I contracted #COVID19 at Mardi Gras in New Orleans. Lost smell, taste, hearing, fever, cough, breathing issues, etc. BUT it seems like they only want to test certain people."
Text K: "We are asking patients who have symptoms of cough, fever, or difficulty breathing to reschedule their non-urgent appointments"
Text L: "Today I was tested for #COVID19 after experiencing a couple days of a bad dry cough, severe congestion, headache, chills/hot flashes, and unmentionable bathroom symptoms... No fever, but apparently that's not a mandatory criteria."

---

**ITERATION 3**
> Top word (headache) is popped from **Q** and processed

---

> Continue until all seed words at **maxDepth** are processed.