# OpenReview forum: "Detecting Emerging Symptoms of COVID-19 using Context-based Twitter Embeddings"
_EMNLP/2020/Workshop/NLP-COVID — NLP-COVID19-EMNLP Poster_

### Official Review · AnonReviewer1 · 2020-09-24
**Original method that deserves to be pursued**

**Rating:** 5
**Confidence:** 3

**Review:**

This work reports an approach to symptom detection on Twitter data. Graph-based relations between a seed word and the more similar words are obtained by means of a similarity measure; the context embedding value is used to represent each word and compute the cosine similarity value with regard to the seed word. The authors applied their method to a collection of COVID-19-related tweets (using the word "cough"), and also conducted their experiment on an ADR corpus (Sarker & González 2015), using the term "pain". Their method obtained similar symtom words in both cases and yielded promising results that could be pursued in other pharmacovigilance tasks.

Strengths:
- I found this is an original method that will interest the audience of the workshop.
- The article is clear, although sone methodological aspects need further clarification and some excerpts minor need revision.

Weaknesses:
- As authors already point out, the method currently detects only single-word terms, not multi-words, which may be essential to detect fine-grained description of symptoms (e.g. "dry cough" vs "wet cough").
- The experiment is only conducted using one word ("cough"); it could provide more solid results if more terms related to the COVID-19 symptomatology had been explored.
- I also missed a pragmatic perspective and brief discussion as to how these types of approaches would be valuable in the real medical-use or pharmacovigilance context. Healthcare professionals could validate these lists of medical terms. This could yield a quality reference dataset, and vice versa, healthcare professionals could get valuable, unknown data about symptoms to take into consideration (and probably not reported to date). The authors could develop this aspect. Namely, developing an interface to explore and show these associated terms would be a great contribution of this work.

The first column in Table 1---"Cough (Manual)"---needs more explanation. Do authors mean that they selected manually a set of words related to "cough", then computed the cosine similarity value? Please, provide more details in the running text.

In Section 2.1, authors have to provide more details about the BERT model they used: e.g. BERT cased or uncase? Did authors use the models released in the paper by Devlin et al. 2019? Did they resort to domain embeddings such as SciBERT or BioBERT? Or did they train their own model? What were the hyperparameters used (learning rate, dimension of hidden state, use of dropout, batch size...)?

- Others (grammar, style...):

    P. 1, Introduction: "SARS-nCOV" -> "SARS-CoV-2" or "SARS-nCoV-2"

    P. 1, Introduction: "pharmacovigilence" -> "pharmacovigilance"

    P. 2, Sect. 2.2: "didn't" -> "did not". Likewise in p. 3. Sect. 2.3: "doesn't" -> "does not"

    P. 2, Fig. 1: colors are not distinguished when printed in black and white; authors should think of an alternative to improve this.

    P. 3, Figure 3, and p. 4, Figure 4: These figures are, in my opinion, small to be read adequately. I suggest authors to make them larger and move one or two to an Appendix.

---

> ### Author Response · Authors · 2020-09-26
> **Thank you for the comments - can easily be addressed.**
>
> We thank the reviewer for their helpful comments, which we believe can be addressed without much effort. Detailed responses follow.
>
> Seed words: In the context of COVID-19, we used three seed words, ‘cough’, ‘fever’, and ‘fatigue’ (Table 2) which were some of the first known symptoms in March 2020. We chose to focus on these as the goal was to find new symptoms over time using the earliest known symptoms. This is especially required for a scenario such as COVID-19, where the pathology of the virus is both unclear and potentially evolving over time - even after several months of the first incidence, newer symptoms are being discovered. We will clarify this in the text.
>
> Practical application: Your suggestion regarding developing an interface is excellent! It would directly tap into the advantage offered by our method to process large volumes of unlabeled data, with minimal labeled data, and output keywords of interest along with the corresponding text. A typical workflow would involve users from the medical/ pharmacovigilance domain inputting a small number of known keywords of interest, detecting the emerging signals (symptoms/ADR) output by the model, and validating them using the corresponding text. This allows for very early detection of symptoms/ADR from social media platforms (as illustrated in Fig 1.) without the need to manually scavenge large corpora. We believe the model’s ability to return corresponding text along with a single-word keyword enables detection of fine-grained contexts (such as dry vs wet cough), thus minimizing its limitation to not detect multi-word terms.
>
> Table 1: "Cough (Manual)" refers to the results obtained from the Manual Context-Text Approach (Section 2.1) while “Cough (Graph model)” refers to the results obtained from Graph-based Iterative Training Approach (Section 2.2). The graph-based approach, as described in the paper, is essentially an iterative and automatic application of the manual approach. The term ‘Manual’ has been used to differentiate it from the graph-based approach, which is more ‘automatic’ and does not require seed text for new seed words that are to be explored. We will clarify this in the text and the caption.
>
> BERT specifics: We finetuned a bert-base-uncased model using the masked language modeling protocol with the following parameters. Num Epochs: 5, Learning Rate: 1e-5, Train Batch Size: 8, Max Sequence Length: 128. Embeddings were extracted from the model by summing the hidden states of the last 4 BERT layers, resulting in 768 dimensional embeddings. HuggingFace library was used for loading pre-trained model weights as well as LM training. Our method is BERT-type agnostic and is easily applicable to other variants like SciBERT, BioBERT, and ClinicalBERT etc. We will open source the code for our method to enable wider application and evaluation.
>
> Grammar, style, etc: Thank you for pointing these - we are able to easily address them.

---

### Official Review · AnonReviewer2 · 2020-09-25
**Good paper and needs further improvement and comparisons**

**Rating:** 5
**Confidence:** 4

**Review:**

The author represented an unsupervised graph-based approach for the detection of symptoms of COVID-19. The paper and idea is good.

Pros
1-The propose method will have several useful applications if it can be generalized and better than traditional methods.

2- The author test their method on 2 different dataset

Cons
1-The methodology needs further explanation
2- There is no comparison with traditional method that are able to capture such symptoms for instance LDA , W2V approaches. Sometime a simple method works better than complex one , particularly when the language context is simple

---

> ### Author Response · Authors · 2020-09-26
> **Thanks for the comment - word2vec and LDA were not effective.**
>
> We thank the reviewer for their comments. We initially experimented with Word2Vec, Glove embeddings, and LDA to find symptoms from tweets. However, we found that the similarity between tokens was the same irrespective of the context in which the word was being used. So with ‘cough’ as the seed word, highly correlated words like ‘fever’ were detected by Word2Vec. However, words like congestion, nausea, diarrhea were not detected by Word2Vec. Consequently, we developed our proposed approach to address this issue. We start with a small set of known keywords and expand the list. Using "similar" words using word2vec doesn't work well due to a lack of context. Our automated method can keep up with evolving symptoms over time. We will clarify these details and also add an appendix with an illustration explaining the iterative graph approach.

---

### Official Review · AnonReviewer3 · 2020-09-25
**A well written concise study presenting an original methodology**

**Rating:** 8
**Confidence:** 4

**Review:**

The paper presents an approach to detecting COVID-19-related symptoms in social media data (Twitter). The authors use word embeddings generated by BERT and build a graph of tokens based on their similarity to the learnt context embedding which allows them to iteratively identify more and more symptoms. The paper would benefit from some technical clarifications and, in my mind, can be extended to the long format.

Strengths:
The study presents an original methodology. Since the method is unsupervised the authors validate its performance by manually calculating precision for a few identified symptoms. They also test the method on a previously developed and annotated dataset for adverse drug reaction detection and show that the method generalises quite well.

Areas for improvement:
- To improve the reproducibility of the study, the details about BERT implementation should be included.
- The process of graph generation is not straightforward and should be accompanied by examples and/or illustrations.
- It is unclear what the similarity scores reported in table 1 represent for manual and graph-based approaches. Are these the cosine similarities between a given token and the seed word “cough”?
- I am not sure what this notation means “##raine”?
- Legends should be added to Figures 2 and 3 to show how node colours correspond to similarity values.
- For the annotated ADR dataset, is it possible to calculate other metrics (e.g. recall, f1, AUC)?

---

> ### Author Response · Authors · 2020-09-26
> **Thanks for your encouragement and comments - an illustration to better explain graph generation process can be easily added**
>
> We thank the reviewer for their encouragement and comments.
>
> BERT parameters: We finetuned a bert-base-uncased model using the masked language modeling protocol with the following parameters: Num Epochs: 5; Learning Rate: 1e-5; Train Batch Size: 8; Max Sequence Length: 128. Embeddings were extracted from the model by summing the hidden states of the last 4 BERT layers, resulting in 768-dimensional embeddings. HuggingFace library was used for loading pre-trained model weights as well as LM training. We will add this to the text.
>
> Graph-based approach: We agree that an illustration would be beneficial to better explain the graph generation process. We have added an example at the end of the response displaying the inputs and outputs of two iterations run with cough as the seed word.
>
> Similarity scores in Table 1 are the average cosine similarities between the word and the seed word “cough”. We will clarify this.
>
> ‘##raine’ is the second half of tokenization of the word ‘migraine’. Words not included in the BERT model vocabulary are broken down into sub-word components by the tokenizer. We also observed that in the case of words not present in the vocabulary, the last half of the word typically gets a higher similarity score. This enables finding keywords of interest that are not directly present in the model vocabulary.
>
> Specifications for Figure 2 and 3: Size indicates the number of instances of the given word with a similarity higher than ‘minSimThreshold’. Color indicates the average similarity score (Cosine similarity) for all instances of the word.
> We will clarify this in the text.
>
> Other metrics for the ADR dataset: We explored this possibility, and however found that there is no definitive set of adverse reactions to act as ground truth. The annotations vary from very specific entities like ‘pain’, ‘headache’ to very broad phrases like ‘a tingling sensation’. This is even more true for COVID-19 as the symptoms are evolving and newer symptoms are being observed even after several months from the first case. We, therefore, did not calculate recall and other metrics that depend on false negatives.
>
> Our approach is intended to work at the intersection of human-machine interaction and provide a ranked list of most important keywords for a user to explore and validate. The model retrieves the most relevant keywords at the top of the list. We will emphasize this aspect in the discussion.
>
> #### **INPUT**
>
> >**Seed word**: cough
>
> >**Seed text**:  [Text A, Text B]
>
> Text A: “I haven't felt "good" for a couple of days. Today I have a headache, cough, chills, fatigued, and feel congested.”
>
> Text B: “COVID-19 can induce intense fatigue and trigger a recurring cough and intermittent fever. It varies from person to person”
>
> >_**cough** is enqueued to ***Q***_
> ---
> #### **ITERATION 1**
> >_Top word (Cough) is popped from ***Q*** and processed \* using user-provided seed text_
>
> \* Processed implies running the similarity search approach where new words are scored based on their cosine distance to the Context Embedding (***CEmb***)
>
> OUTPUT 1
>
> * fever, [Text C, Text D]
>
> * headache, [Text E, Text F]
>
> * throat, [Text G, Text H]
>
> Text C: “Due to #COVID19 concerns, all visitors are being screened for symptoms such as cough, runny nose, fever and difficulty breathing”
>
> Text E: “I have a dry cough, chest pain and feeling lethargic as hell plus a headache. Not to mention feeling like I have to take deep breaths. Went to urgent care and they basically wrote it off as a UPI because COVID-19 tests are unavailable.“
>
> Text G: “I'm still really sick. Been coughing so much my ribs are sore. Have a sore throat, am achy, congested, dizzy, nauseated.”
>
> >_**fever**, **headache** and **throat** are enqueued to ***Q***_
>
> >_***CEmb*** is updated_
> ----
> #### **ITERATION 2**
>
> _Top word (fever) is popped from ***Q*** and processed. Text corresponding to ‘fever’ identified from previous iteration (Text C, D) is used as seed text\**_
>
> ** This allows for the automatic evaluation of new seed words without the need for manual seed text input
>
> OUTPUT 2:
>
> * taste, [Text I, Text J]
>
> * breathing, [Text K, Text L]
>
> * congestion, [Text M, Text N]
>
> Text I: “Pretty sure I contracted #COVID19 at Mardi Gras in New Orleans. Lost smell, taste, hearing, fever, cough, breathing issues, etc. BUT it seems like they only want to test certain people.”
>
> Text K: “We are asking patients who have symptoms of cough, fever, or difficulty breathing to reschedule their non-urgent appointments”
>
> Text L: “Today I was tested for #COVID19 after experiencing a couple days of a bad dry cough, severe congestion, headache, chills/hot flashes, and unmentionable bathroom symptoms... No fever, but apparently that’s not a mandatory criteria.”
>
> >_**vomiting**, **breathing** and **congestion** are enqueued to ***Q***_
>
> >_***CEmb*** is updated_
> ----
> #### **ITERATION 3**
> >Top word (headache) is popped from ***Q*** and processed
>
> >Continue until all seed words at ***MaxDepth*** are processed.